# Governance Structure of Rural Homestead Transfer in China: Government and/or Market?

Yongchao Zhang [1,2,*](image) , André Torre [3](image) and Marianne Ehrlich [3]

1   Department of Public Management, Nanjing Agriculture University, Nanjing 210095, China
2   China Land Problem Research Center, Nanjing 210095, China
3   UMR SAD-APT, University Paris-Saclay, INRAE, AgroParisTech, 75005 Paris, France;
    torre@agroparisteach.fr (A.T.); marianne.ehrlich@agroparistech.fr (M.E.)
*   Correspondence: 2016209023@njau.edu.cn

**Abstract:** Rapid urbanization in China has triggered the mass migration of rural populations to cities. These policies have resulted in a shortage of construction land for the urban population and in an inefficient use of rural homestead, causing a tremendous waste of rural land resources. Rural homestead transfer has been identified as a tool to solve this problem: the saved construction land can be reclaimed to cultivated land, and the construction rights are transferred to urban areas, where they can be used to build new households for the demanding population. We consider that transaction costs analysis can help understand the governance structures of the homestead processes, and provide avenues for further research and policy recommendations. Our article draws on the practices and experiences gained in three areas in rural China (*Jinzhai*, *Yiwu*, and *Jiangning* in the Anhui, Zhejiang, and Jiangsu Provinces, China). Based on the empirical cases and information produced from the villages and households survey, we developed an analytical framework of transaction cost. Results suggest that *Yiwu* has lower assets specificity, lower uncertainty, and transaction frequency, hence the market driven model played a major role. In *Jinzhai*, which presents higher assets specificity, higher uncertainty, and transaction frequency, the government-oriented model also played a major role. In *Jiangning*, where most transaction attributes are situated in the middle, the mixed-oriented model acted as a main role in the local area. Our conclusion shows that various governance structures are appropriate for different areas of rural China, which is significantly related to the local transaction attributes. We suggest promoting the governance structure which corresponds to the local resource endowment, human and residential condition, the cultural atmosphere, and also the institutional (official) atmosphere. Matched with the government or/and market governance structure, it may develop the resource allocation efficiency and improve economic performance.

**Keywords:** homestead transfer; transaction cost; governance structure; *Jinzhai*; *Yiwu*; *Jiangning*

## 1. Introduction

Like in any other country, rapid urbanization in China has triggered the mass migration of rural populations to cities. The Chinese central government launched different policies of regulation to try to mitigate the land uses, like the quota for the construction land granted every year to the city, or the policy of "cultivated land gross dynamic balance" (The goal of this policy is to maintain the stable amount of cultivable land in the country. If the urban construction leads to an increase of land use in the suburbs of the city, a corresponding amount of construction land must be re-cultivated in the countryside.), which strictly controls the growth of construction land and balanced it with the development of cultivated areas. Due to these policies, it resulted in a shortage of construction land for the urban population and in an inefficient use of rural homestead [1], with the phenomenon of "hollowing villages" and vacant residential houses, causing a tremendous waste of rural land resources [2,3]. This contradiction between land supply and demand has become

the dilemma of increasing urbanization and industrialization, with a great demand for construction land in the rapid economic development areas [4].

Rural land is generally classified into three types: agriculture land, collective construction land, and unutilized land. Rural collective construction land can further be divided into commercial land, public land, and rural homesteads, which account for the largest share [5,6]. Rural homestead transfer (RHT) has been identified as a tool to solve this problem in cities and towns [5,7–9]. The homestead can be directly rented or transferred to different subjects, like households, developers, and enterprises, or indirectly turned into construction land indexes and be traded on the market. The saved construction land can be reclaimed to cultivated land, and the construction rights are transferred to urban areas. For a long time, the legislation of China was quite strict. It treated the homestead as a residential land for households, distributed to farmers as a benefit and not transferable to the market [10,11].

Nowadays, the government is urgently demanding innovations regarding this issue. Thirty-three pilot areas for the land system reform in China were chosen in 2015, experimenting different types of RHT practices. (1) Spontaneous market transfer, named as "market driven model". (2) "Government oriented model" [12–15], based on respective increase and decrease of urban and rural construction land in order to achieve a more balanced state. (3) "Government and market mixture-initiated model" [13,16], where the government negotiates with the farmers and regains their homestead, and then does redevelopment, sometimes involving several local or external firms [17]. Scholars have tried to summarize this diversity of situations according to three main types of RHT: division between government oriented, collective-led, and farmers-led homestead transfer [15]; developed and underdeveloped (In China, generally speaking, according to the local economic development level, it can be divided into two types: the developed areas in the east of China, and the undeveloped areas in the vast midwest) areas transfer models according to the local economic development level [18]; and distinction between legal and illegal transfers [19]. However, these classifications lack theoretical basis and without criteria for support, they present limitations for a comprehensive representation of different models of RHT and their characteristics.

In line with a few scholars [20–24], we consider that transaction costs analysis can help in understanding the governance structures of the homestead processes [3,12,13], and provide avenues for further research on their formation mechanism, more specifically, the subdivision of various complex homestead transfer patterns into different stages. Our article draws on the surveys and experiences gained in two pilot areas (*Jinzhai*, Anhui Province, and *Yiwu*, Zhejiang Province) and one area which is not included in the pilot sites list (*Jiangning*, Jiangsu Province). These areas are respectively settled in advanced, developing, and underdeveloped regions but their experiences in terms of homestead transfer prove to be successful.

We assume that the three models of transaction process and homestead transfer—government oriented, market driven, and mixed initiated models [12,24–26]—mainly correspond to the replacement of homestead in *Jinzhai*, to the "land coupon" in *Yiwu*, and to the "homestead renting model" in *Jiangning*. Based on this experience, the goal of the article is to define a suitable classification of governance structures for rural homestead transfer, and to find possible improvement plans to choose the suitable governance structure for each type of area. The insights can provide an input to the design of appropriate governance structures of rural homestead transfer, and improve the efficiency of resources allocation.

The remainder of the paper is organized as follows: Section 2 presents the theoretical approach and the research hypotheses in terms of transactions costs and attributes; Section 3 provides information about the methodology and the empirical analysis; Section 4 presents the empirical results for the three different cases; Section 5 discusses the hypotheses and the comparison between the theoretical and actual governance structures of homestead transfer, and Section 6 presents the conclusion and policy implications.

## 2. Theoretical Analysis and Research Hypotheses

The ongoing and sustainable land use policy and new RHT methods and techniques are aimed at diversifying the household livelihoods and improving their income status in China. Most of the previous literature on homestead transfer use transaction costs and governance structures approach [12,16,27,28]. We will base our approach on this *New Institutional Economics* [29–32], which explicitly integrates the diversity of organizational forms (or governance structures) to govern the transfer of goods and property rights.

### 2.1. Transaction Costs Analysis and Its Application to Homestead Transfer

The homesteads transfer is a combination of a series of transactions: it may include the transaction between farmers and government for giving up homestead for compensation, the issues of the homestead index transaction between government (collective village) and land users (developers), the transaction issues of the use of homestead (indicators) and the resettlement of farmers. As the RHT is the aggregation of a series of transactions, it is necessary to divide it into clear and accurate stages, and choose suitable governance structures for different stages. For a given transaction (with given transactional attributes), the transaction costs theory suggests that one governance structure will outperform the alternatives. This is the "alignment principle" described in Williamson [29–31].

Some scholars, on the basis of their observations about the case of RHT in China, have discussed the transaction costs hypothesis and the choices of different contract forms. They divided the general transaction into three different stages: the fees of information searching (before the transaction), the fees of communication and negotiation (during the transaction), and the fees of enforcement and supervision (after the transaction) [20–22]. Tang identified 83 procedures regarding the non-agricultural land use in rural areas and utilized this approach to find the suitable governance structure for each procedure [12]. Others have analyzed the transaction costs in different stages of farmland mortgage, and designed an appropriate governance structure for each stage, dividing them into governance chains and mixed governance structure [13].

We follow these approaches and propose an analysis of RHT on the basis of transaction cost approach. Building on the fact that economic agents have the choice between alternative governance structure, we assess the suitability of different governance structure (market system, mixed initiated system, and enterprise system) for the homestead transfers, considering that the institutional environment [33] remains unchanged. The micro transactions entail a specific transfer of rights, services, and information, making it necessary to divide the homestead transfer into multiple stages for inspection.

A transaction is determined by the comprehensive consideration of the production and transaction costs. Asset specificity is the significant condition for the existence of transaction costs [34,35], that is, without considering the uncertainty of transaction, the best choice for governance structure is determined by the asset specificity. As a result, the standard contract is the classical contract, accomplished by the market (market driven structure). The transactions with high degree of assets specificity, transaction frequency, and uncertainty belong to some kind of relational contract, accomplished by bureaucracy for centralized management [22,33,36,37]. The transactions that are situated between the two sides are suitable to be organized by the so-called mixed form.

### 2.2. The Indicators

Transactions are linked with three main measure indicators: the assets specificity, the uncertainty of transaction, and the transaction frequency.

#### 2.2.1. Asset Specificity

Asset specificity refers to the degree to which an asset can be redeployed to alternative uses and by alternative users without sacrifice of productive value [30]. It includes various types, such as site specificity (location factors), physical, human, and dedicated brand assets specificity [13,20,21,31,36]. As for the local assets specificity of homestead we choose

their location condition. As for the physical asset specificity, we choose the cultivated land resource endowment of the area, which can describe their reliance as well as their values for the farmers. Off-farm work level indicates the human assets specificity.

Homestead represents a key property for households. It is their residence but it also has the function of storing agricultural mechanization and cereal products, and even the function of breeding livestock. The degree of its asset specificity is determined by the degree to which farmers depend on the rural production and living environment. If the farmers are mainly engaged in agriculture, forestry, fishing, and other agricultural production activities, they present a high specialization and low generally recognized value, which is difficult to convert to other uses when farmers exit from the rural environment. Otherwise, if the farmers take off-farm work as the main business, and thus rely on the rural production and living environment to a lesser extent, the homestead present a low specificity degree. It may result in minor loss when the farmers leave the environment on which they rely [16].

In addition, the transaction may involve different property topics. When the transaction covers multiple villages, the property rights will be more heterogeneous; while, if the homestead transfer only refers to one village, the property rights is less heterogeneous. The more heterogeneous the property rights of the transferred homestead, the stronger the assets specificity. On the other hand, the larger the scale of the transferred homestead and the larger population of farmers' participation, which will result in the great impacts on the period and progress of the transaction, accordingly, the higher the assets specificity. Hence, we take them into account and added the heterogeneity of property rights and the scale of transfer to the indicators-system we use for measuring the assets specificity.

Overall, among them, the location of the homestead belongs to the geographical asset specificity. Cultivated land resource condition belongs to physical assets specificity. Social security function of homestead and property function of homestead belongs to brand assets specificity. Heterogeneity of property rights and scale of transfer belongs to special/dedicated asset specificity. Off-farm work level belongs to human asset specificity.

### 2.2.2. Uncertainty of Transaction

Williamson has referred to two types of uncertainty [29]. One is the predictableness of the environment, which varies according to technological innovation, markets volatility, and so on. The other one is the behavioral uncertainty, such as the uncertainty arising from missing or asymmetric information, which means possible opportunistic behavior. The bigger the uncertainty, the greater the transaction cost.

In the process of RHT, the local government interviewed the household door to door, organized discussions with the representatives, seeking opinions, organizing public hearings, and advertising the transfer scheme and measures according to the voluntary compensation principles. However, the level of enforcement and situation of farmers' participation is quite different in various areas. The exchange between the farmers and the government can be insufficient, and the information asymmetric. Hence, we choose the uncertainty of information acquisition as an indicator to measure the uncertainty of the environment, the uncertainty of corruption in the government department or the collective organization [27], used for measuring the uncertainty of behavior, so as to observe its effects on the transaction costs of homestead transfer.

### 2.2.3. Transaction Frequency

Frequency of a transaction refers to the number of times a given transaction is repeated. According to Williamson, "the cost of specialized governance structures will be easier to recover for large transactions of a recurring kind" [29–31]. In other words, for given levels of asset specificity, the greater the volume of trade, the more likely the benefits of hierarchical governance. This means the government-oriented model may be more suitable for the more frequent transactions. According to the different transaction frequencies, we identified disposable, occasional, and multiple transactions. In our different case studies, the farmers proceed homestead transfer for only once. After the realization of

centralized residence, they could no longer move back to the original homestead. Their homestead transfer belongs to disposable transaction [16]. Therefore, the differentiated transaction frequency will have inconsistent influences on the transaction cost of different homestead transfer patterns. The more the transaction frequency, the higher the transaction costs [12,27]. The specific variables and their influences please see Table 1.

**Table 1.** The transaction attributes and their application to homestead transfer.

| Dimensions | | The Transaction Attributes | Application to Homestead Transfer |
|---|---|---|---|
| **Asset specificity** | Geographical assets specificity | Local assets specificity | The more specific the location of the homestead, the higher the assets specificity |
| | Special/dedicated asset specificity | Heterogeneity of property rights | The greater the number of villages involved in the homestead transfer project, the more heterogeneous the property subjects, the stronger the assets specificity |
| | | Scale of transfer | The greater the volume of land area involved in homestead transfer projects, the larger the scale and the population of farmers, the higher the assets specificity |
| | Physical assets specificity | Cultivated land resource condition | The more cultivated land resource, the higher the assets specificity |
| | Brand assets specificity | Social security function of homestead | The stronger the social security function, the higher the assets specificity |
| | | Property function of homestead | The higher the market value of the assessed homestead, the stronger the property function of homestead, the lower the assets specificity |
| | Human asset specificity | Off-farm work level | The higher level of off-farm work, the lower the assets specificity |
| **Transaction uncertainty** | | Uncertainty of information acquisition | The bigger the uncertainty of information acquisition, the higher the transaction costs |
| | | Uncertainty of corruption | The bigger the uncertainty of corruption, the higher the transaction costs |
| **Transaction frequency** | | Transaction frequency | The higher of transaction frequency of homestead transfer, the higher the transaction costs |

### 2.3. The Main Hypotheses

Through observations of typical pilot areas, we established a set of index systems, which reflect the differences of transaction costs among three different stages for each case study. Because each model of homestead transfer contains complex procedures, we put forward three hypotheses, which are listed below, for different cases of RHT and every stage of the RHT model.

**Hypothese 1 (H1).** *When the assets specificity is higher, the uncertainty and the transaction frequency are higher. Then, the government-oriented governance structure will be more suitable.*

**Hypothese 2 (H2).** *When the assets specificity is lower, the uncertainty and the transaction frequency are lower. Then, the market-driven governance structure will be more suitable.*

**Hypothese 3 (H3).** *When the asset specificity situates middle level, the uncertainty and the transaction frequency situate at relatively medium level. Then, the mixed-initiated governance structure will be more appropriate.*

### 3. Methodology and Empirical Analysis

In 2015, the Chinese government chose 33 pilot sites for the land system reforming. To make the investigation of the diverse forms of possible homestead transfer, we selected three case villages: (a) one in a region characterized by an advanced economy (*Yiwu*, Zhejiang province); (b) the second one in a region characterized by a developing economy (*Huanglonxian*, *Jiangning* region, Nanjing city, Jiangsu province); (c) the last one located in a region with an underdeveloped economy (*Jinzhai*, Anhui province). We selected them because of the differences in regional economic conditions, the typical local practices, and most of all because they happened to represent the three different governance options that we want to explore. The homestead transfer in the *Yiwu* case is market driven, in *Jinzhai*, by the government promoted force, while in the *Jiangning* case, it is a mixture of market and government intervention.

The population of *Yiwu* is 953.312 inhabitants, the urbanization reached 76.2%, and the GDP per capita 154,242 Yuan in 2018 and 15.97 billion USD in 2016, belonging to China's top 100 counties for business environment released in 2019. The percentage ratios of gross product of agriculture, industry and service are respectively of 1.7%, 32.8%, and 65.5% [38]. The registered population of *Jinzhai* is 683.000 inhabitants; the urbanization rate of registered population is 16.3%, and the county's GDP per capita was USD 3144 in 2018 [39]. The structure ratio of three sectors (agriculture, industry, and services) is respectively of 20.2%, 38.8%, and 41.0%. The permanent residents of *Jiangning* is 1,248,500, the urbanization rate of registered population is 73.1%, and the district GDP per capita was USD 25,843 in 2017. The percentage of the structure ratio of three industries (agriculture, industry, and services) adjusted to 3.2%, 53.3%, and 43.5% [40]. The more specific information can be found in Appendix A.

In order to obtain suitable comparable information, we conducted unstructured interviews in January 2018, a powerful qualitative research method that prioritizes the validity and depth of the interviewees' answers. First, initial interviews with the officials from the Bureau of Land resources of local counties were carried out in order to better understand the RHT procedures and the potential impacts it can have from the perspective of the different levels of government. Then followed interactive interviews with the farmers affected by the RHT projects, thus the broad data covering the socio-economic conditions have been collected. The village head was interviewed to analyze the restructuring procedures and the participation status from these projects. The interview contents included the fundamental situation of the studied territories, the resource condition, and the general situation about the project of "homestead transfer". Our findings are based on detailed texts, audio recordings, and policy supply documents of local government departments and township brochures.

### 3.1. The "Land Coupon" Model of the "Street of West City" in YIWU

*Yiwu* City carried out homestead system reform in 2015 and launched a "land coupon" model, which is famous throughout the country. Transaction is carried out publicly through a property exchange platform which is declared on the website of the government, and then the county-government proceeds public bidding in the Bureau of natural resources, the different demand subjects carry out the competition for the "land coupon".

In the first stage, the farmers need to apply for the consolidation of their homestead towards the official department of township (or sub-district office) and submit the application form and the documents proving ownership of the house. They will choose voluntary withdrawn from their homestead on the perquisite that they should have another stable residence (like houses in cities). Only under these conditions they can choose to transfer their homestead, and the withdrawn homestead will be reclaimed into arable land, which will need to be inspected and checked by the BLR (land resources bureau) of *Yiwu*. The "land coupon" will be granted to the farmers when the reclaimed arable land has met the required standard. Since, even though there exist the possibilities that the authorities may have corruption behaviors in the process of the inspection for the quality of the re-

claimed arable land, the space for "rent-seeking" is not very large. The newly generated "land coupon" belongs to the farmers. The next step happens at the property rights trading platform, which may include both initial and secondary transactions. The bureau of land resources ascertains and publishes its estimated prices according to the acquisition costs. The first transaction value of the "land coupon" should not be lower than the indicated price. However, secondary trading will not be subject to the guide price. All the demand sides participated in the competition though the forms of public auction. After the "land coupon" has been traded, the demand sides may search for the plots they need on the market. The municipal government is responsible for coordinating and arranging the use of the "land coupon" for various projects and handles the authority for approval of farmland transformation. It may proceed imposing, and then selling the land in the procedure of public bidding, auction, and listing. All the enterprises that get the "land coupon" will compete for the plots. Based on a rural property transaction center, the RHT is conducted in the form of competitive tender in the market, even if the demand sides come from different industries and departments. Therefore, this case can be named a market driven system.

*3.2. The Transfer Model of Homestead Long-Distance Replacement in Jinzhai*

*Jinzhai* belongs to the model of long-distance removal for poverty relief. Our case presents long-distance homestead replacement in *Baita* Township, *Jinzhai* County. This process can be divided into three stages from the point of view of the local town government. In the first stage, it has to evaluate the amount of compensation to provide to the farmers, in order to promote their removal from original housing. The government needs to pay a great deal of expenses to understand the farmers' will, to encourage the households to transfer their homestead, and to make reasonable compensation in the form of currency and houses/apartments as well. In the second stage, the government needs to proceed to site selection and planning placement area for the households, and may need to acquire land in the placement area, submit to the higher authority for approval. A part of these homestead can be used for the construction of concentrated residential community, the rest of which is transformed into cultivated land. On that basis is built a construction land index, used to create rights of construction. At the last stage, the county government sells the land development rights to urban areas suffering from an urgent need of construction land. The previous scattered households in the villages are placed together in a concentrated residential community, in which the living conditions have been improved. In that case, the government has taken a leading role in the process of long-distance replacement. Therefore, it can be named the government-oriented model.

*3.3. The Transfer Model of "Homestead Renting" in Huanglonxian, Jiangning Area, Nanjing*

In the RHT process of *Huanglonxian*, *Jiangning* area, both the government and market have played an important role.

In the first stage of the government-oriented model, the government needs to play the role of the advertisement and to persuade the households to transfer their homestead, in order to proceed to renovation. They encourage and guide the households to participate in the reconstruction of the village. In the second stage, the government packages the households' homestead and then transfers them to an enterprise named "Development & construction Co., Ltd." uniformly. The company signs a contract with the government or the collective economic organization, and then pays for the rentals. A part of homesteads is used for the development of village tourism, tea house, and the ecological park by the enterprise. Owing to the establishment by the government of an intermediary agent between the enterprise and the households, the transaction procedures have been simplified.

On the other hand, in the first stage of the market spontaneously renting, the farmer who has transferred demand publishes the transfer information, as well as the farmer who has the operational requirements. The trading parties both search for the transfer objects and publish the transfer price. In the second stage, the two parties sign the contract; make an agreement about the price and the deadline, during which there exists a price

and competition mechanism. The homesteads are transferred among the households, which are used for the farmhouses and guest-rooms management by the households themselves. The government does not intervene, or only to improve the public service facilities, the execution and maintenance of the contract. The government mobilized the households to transfer their homestead and then proceed to reconstruction, afterwards renting the saved construction land to the enterprises. Meanwhile, there are also spontaneous transactions among the households. Therefore, this model is a combination of government and market systems.

## 4. Results

In our results, we use transaction costs analysis to understand the governance structures of the homestead processes [20–23], and more specifically, the subdivision of various complex homestead transfer patterns into different stages. We base our analysis on the fact that transaction cost attributes can be divided into three different dimensions: assets specificity, transaction uncertainty, and transaction frequency.

### 4.1. The Transfer Model of Yiwu: Market and Low Transaction Costs

In *Yiwu*, due to low geographical, physical, and brand assets specificity, information searching costs are low. The "street of west city" is located in the outskirt of *Yiwu* city, which belongs to the most developed area in east of China, a favorable location and transport condition, thanks to the proximity of Hangzhou, the capital of Zhejiang Province, and Shanghai. Hence, the geographical assets specificity is low. Owing to the favorable location conditions, *Yiwu* has beneficiated from the driving effect of the central cities. The property function of the homestead has been prominent; its assets value increased gradually, which induced the tremendous residence demand of workers who are working in the surrounding areas. Meanwhile, it produced an effective incentive for numerous capitals and enterprises. Therefore, the brand assets specificity is low. The farmland resource is quite barren, a great number of cultivated lands have been transformed into non-farm constructed land due to the urbanization process, per capita cultivated land resource is fairly scarce, agricultural mechanization as well. This indicates that the physical assets specificity is also relatively low.

The cost for negotiation and signature of contract is low. The degree of education of the native rural labors is relatively high, which may enhance the native off-farm work level to a large extent. In the second place, the location nearby big cities make it easier and convenient for the farmers to find a job and solve the employment problems. So, the human assets specificity is low. The heterogeneity of property rights of transacted homestead is quite strong as they belong to different subjects within a collective organization. Meanwhile, the transfer scale is relatively large. Thus, the "dedicated assets specificity "is low. Therefore, due to the lower human asset specificity and special/dedicated asset specificity, biding and singing up are easier and smooth, as well as the fees related to the course of the competitive tender.

Due to a low uncertainty of information and behavior, and transaction frequency, the cost for maintenance and execution of transactions is also low. Local township government and collective local organizations like village leaders have done great publicity and advertisements for the RHT, inside and outside the area. Due to the fairly unobstructed information, the uncertainty of the environment is low. Owing to the reputation cost of corruption in the process of homestead transfer, the government and collective organization are not willing to take a risk to proceed to rent-seeking. As a result, the uncertainty of behavior is relatively low (see Figure 1). As for transaction frequency, the "land coupon" can be accomplished only once.

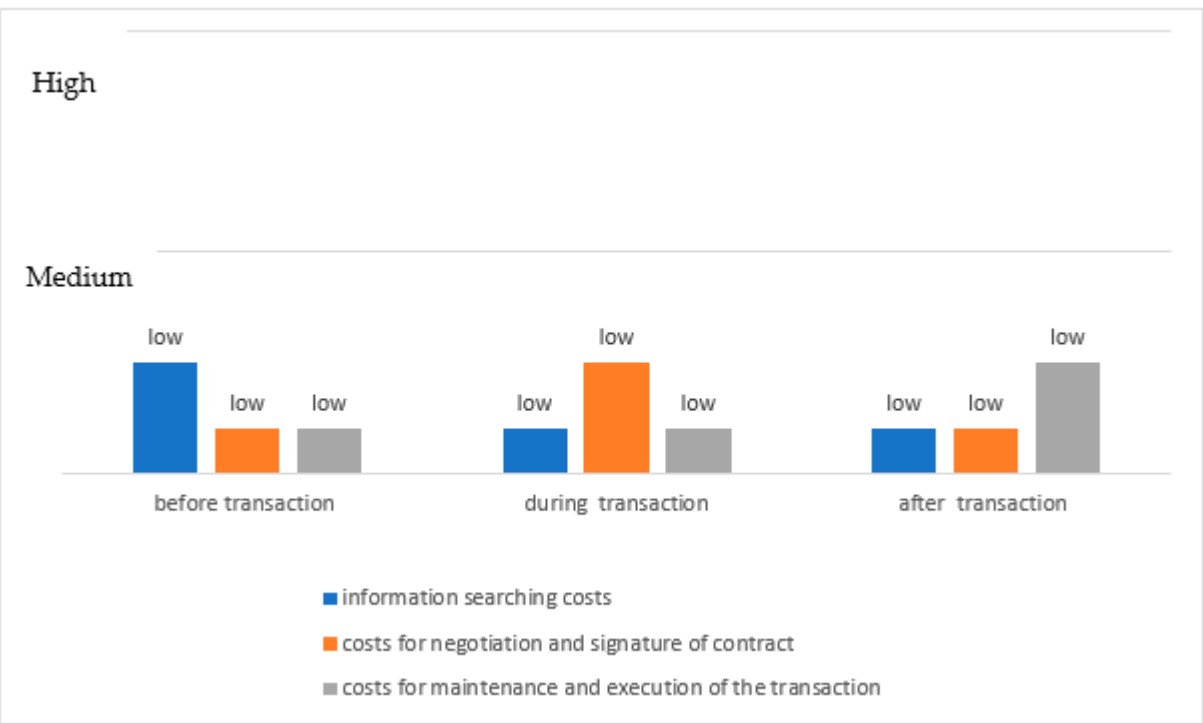

**Figure 1.** Transaction costs at different stages in the "land coupon" model.

*4.2. The Transfer Model of Homestead Long-Distance Replacement in Jinzhai: Government and High Transaction Costs*

In *Jinzhai*, due to high geographical, physical, and brand assets specificity, information searching costs are high. *Jinzhai* is located in the remote outskirts of midwest China, quite far from the central city, the location condition is not so good. Due to the poor transport infrastructure, it is difficult to reach and interact with other parts of China. Therefore, the geographical assets specificity is comparatively high. Secondly, homesteads in the remote outskirts of the city that cannot beneficiate from the externalities of the urban areas still provide service functions for agricultural production (drying place and storage of agricultural mechanization), and do not present favorable conditions for introducing internal and external enterprises. Therefore, the brand assets specificity is comparatively high. The farmland resources are fairly abundant, the farmers conduct agricultural production and related activities that allow for a sustainable livelihood, which indicates that the physical assets specificity is comparatively high.

The cost for negotiation and signature of contract is high as well. The low level of economic development and overall education of farmers have restricted the off-farm employment level to a great extent. Consequently, the human assets specificity is high. Furthermore, the heterogeneity of property rights is strong, related to the great number and types of homestead, and the diversity of farmers behaviors. The number of homesteads participated in the transfer is large, the government needed to mobilize numerous households to participate in the removal and interviewed them one by one in order to build information about the property rights. Thus, the dedicated assets specificity is high.

Due to a high uncertainty of information and behavior, the cost for maintenance and execution of transactions is high. Owing to the inadequate broadcast about RHT of the government and the collective organization, the narrow channel of obtaining information and the reduced opportunity to communicate with the outside, the information about the RHT is comparatively shortcoming. Due to the nonexistence of an effective supervision mechanism, the supervision expenses of corruption are high. As a result, the uncertainty of behavior is fairly high (see Figure 2). As for transaction frequency, the long-distance replacement can be accomplished just for once.

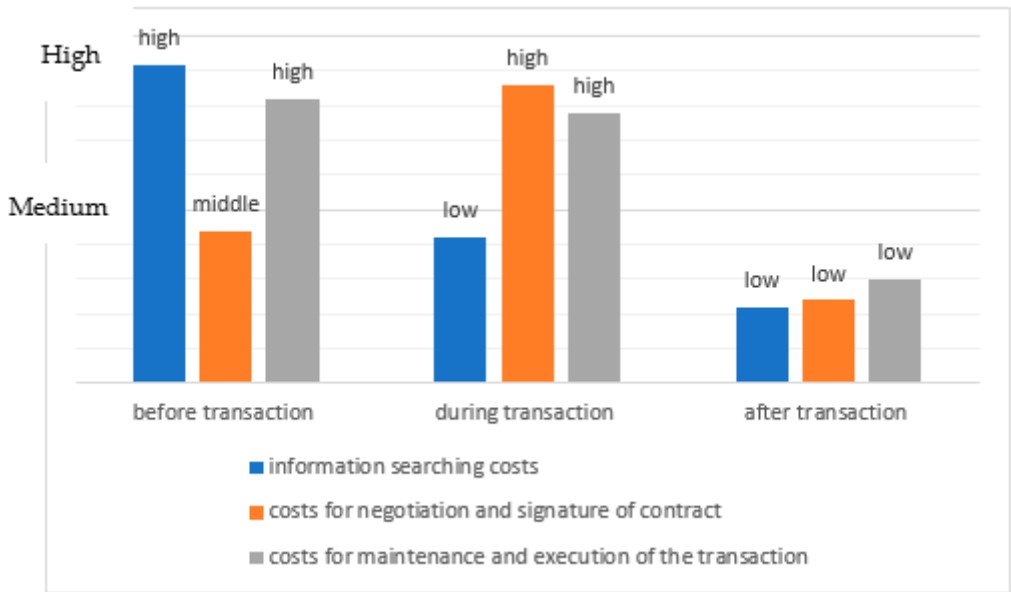

**Figure 2.** The transaction costs at different stages in the "homestead replacement" model in *Jinzhai*.

*4.3. The Transfer Model of "Homestead Renting" in Huanglonxian, Jiangning Area: High Searching Costs*

In *Jiangning*, due to the medium level of geographical, physical, and brand assets specificity, information searching costs are medium. *Jiangning* is located in a relatively developed area in the east of China and at a short distance from Nanjing and Ma' Anshan cities. If we make a comparison with the regional advantages of *Yiwu*, which is located near the metropolis Hangzhou, the geographical assets specificity appears situated in the middle level. Secondly, the location in the suburban area of Nanjing city is beneficial in terms of tourism and service functions. Owing to the great number of amenities and beautiful and pleasant natural sceneries, and also the cultural and recreational facilities, homesteads gradually highlighted and attracted a large number of tourists, promoting travel and local consumption. At the same time, it has formed effective incentives for social capitals and enterprises. It is quite common to find homesteads rented and farmhouses set up as guest-rooms. However, the property function of homesteads is still insufficient, so the brand assets specificity is situated in the middle level. Furthermore, the cultivated land resources are comparatively barren, the majority of the farmland has been converted into non-farming construction land owing to the urbanization process, and the agriculture instruments are also insufficient, but a little bit more important than that of *Yiwu*, which indicates that the physical assets specificity is situated in the middle level as well.

The cost for negotiation and signature of contract is medium as well. Due to the proximity of the cities, the proportion of local off-farm employment is fairly higher than that of migration. The education degree of the local labor force is relatively high, which also promoted the local off-farm employment level to a certain degree, yet still lower than that of *Yiwu*. Therefore, the human assets specificity lies in the middle level. Meanwhile, the heterogeneity of property rights is stronger, because of the multiple transactions of homesteads from different property right subjects. The transfer scale is also big, involving numerous households. However, it is only fruitful for the farmers, with a higher degree of non-agriculturalization in local area, but much smaller than in *Jinzhai*, thus the dedicated assets specificity lies in the middle.

Due to a high uncertainty of information and behavior, the cost for maintenance and execution of transactions is medium. The government's full publicity of the RHT policy, as well as the disclosure in the collective affairs, increased the information channels reaching the farmers, as well as the opportunities to communicate with the outside. However, it is not

unobstructed and as convenient as that of *Yiwu*, therefore, the uncertainty of environment lies in the middle level. We cannot compare the credibility and supervision mechanism with *Yiwu*, which may result in corruption behaviors, while the possibility is quite lower than that of *Jinzhai*. Therefore, the uncertainty of behavior also lies in the middle level (see Figure 3). As for transaction frequency, in the process of homestead renting, it may need one or multiple times to accomplish the transactions.

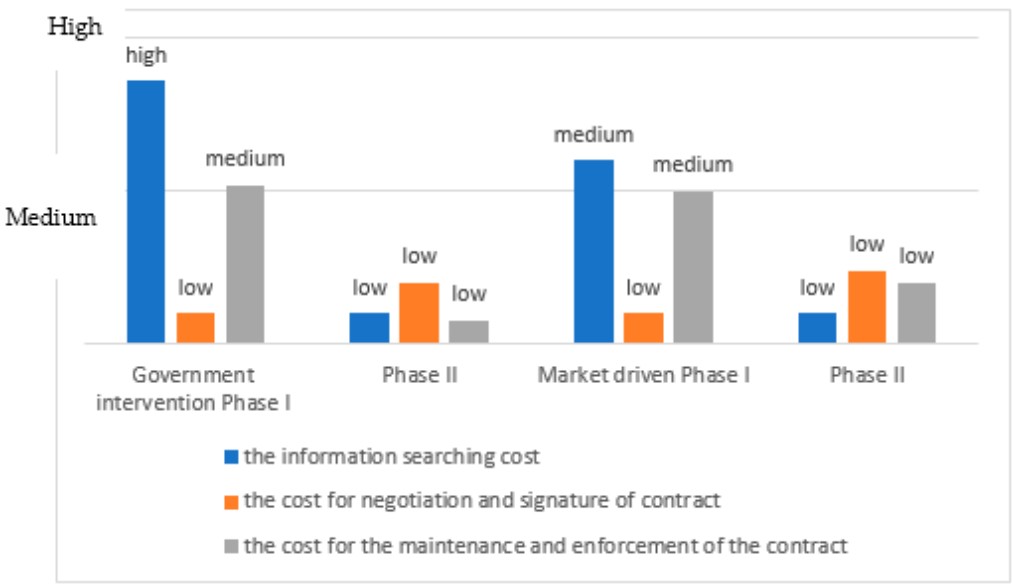

**Figure 3.** The transaction costs at different stages in the "homestead renting" model of *Jiangning*.

### 4.4. General Assessment

The main results of the RHT processes in the three case studies can be summarized in Table 2.

**Table 2.** The governance structure and the transaction attributes in the three case studies.

| Dimensions | The Transaction Attributes | "Land Coupon" of *Yiwu* | Homestead Replacement of *Jinzhai* | Homestead Renting of *Jiangning* |
|---|---|---|---|---|
| **Asset specificity** | Heterogeneity of property rights | weak | strong | strong |
| | Local assets specificity | low | strong | moderate |
| | Scale of transfer | big | big | big |
| | Cultivated land resource condition | low | high | high |
| | Social security function of homestead | weak | high | high |
| | Property function of homestead | strong | high | high |
| | Off-farm work level | high | high | low |
| **Uncertainty of transaction** | Uncertainty of information acquisition | strong | strong | moderate |
| | Uncertainty of corruption | low | weak | moderate |
| **Transaction frequency** | | low | weak | moderate |
| **Expected transaction costs according to the transaction cost theory** | 10 variables (7 for assets, 2 for uncertainty of transaction, 1 for transaction frequency) | - -+- -+++- - | ++++++++- - | +0++++-000 |
| **Transaction costs** | | low | high | middle |
| **Governance structure** | | market system | government system | mixed system |

Source: Data research (Note: "+" means high or strong, "0" means middle or moderate level, "-" means low or weaker level. Each sign represents one type of transaction attributes).

## 5. Discussion

As revealed by the previous developments, the homestead transfers in *Yiwu*, *Jinzhai*, and *Jiangning* belong respectively to the market driven, government oriented, and mixed initiated models. But do these transaction attributes match with the governance structure and therefore minimize the transaction costs effectively?

### 5.1. Return to the Hypotheses

Since the transaction attributes of each dimension that characterize asset specificity are relatively high/strong, *Jinzhai*'s overall asset specificity and transaction attributes of uncertainty are relatively high. Although the transaction frequency is low, the total transaction cost is relatively high. *Jinzhai*'s homestead replacement model belongs to a government-oriented governance structure. Our Hypothesis 1 is verified.

In *Yiwu*, the transaction attributes of all dimensions that characterize asset specificity are relatively low/weak. *Yiwu*'s entire asset specificity, transaction attributes of uncertainty, and transaction frequency are relatively low. Therefore, the total transaction cost is relatively low. *Yiwu*'s "land coupon" model belongs to a market-driven governance structure. Our Hypothesis 2 is verified.

In *Jiangning*, the transaction attributes of all dimensions that characterize asset specificity, uncertainty, and the transaction frequency are all situated at the relatively medium level. Therefore, the total transaction cost is relatively medium. *Jiangning*'s "homestead renting" model belongs to a mixed-initiated governance structure. Our Hypothesis 3 is verified.

### 5.2. Comparison between Expected and Actual Governance Structures

In order to open a way for discussion and possible improvements of the actual situation, we can assess the correspondence between the observed types of governance structure for each form and stage of homestead transfer results and the theoretical corresponding situations. Transaction costs approach defines the appropriate type of governance structure given the characteristics of the main attributes. Namely, high assets specificity, uncertainty, and transaction frequency are associated to government-oriented structure, low assets specificity, uncertainty, and transaction frequency to market driven structure, and medium transaction attributes to a mixed-initiated by both the government and the market model. Our research reveals that the observed results are slightly different (see Table 3).

**Table 3.** Practical and matching governance structures in the different areas.

| Governance Structure of *Yiwu* RHT | The Generated Stage of "Land Coupon" | | The Transaction Stage | The Floor Stage of "Land Coupon" |
|---|---|---|---|---|
| Observed | Market system | | Market-system | Government-system |
| Theoretical prediction | Market system | | Market-system | Market-system |
| Governance structure of *Jinzhai* RHT | Before replacement | | Replacement | After replacement |
| Observed | Government system | | Government system | Government system |
| Theoretical prediction | Government system | | Government system | Market-system |
| Governance structure of *Jiangning* RHT | First stage of government intervention | Government intervention | First stage of spontaneous transfer | Spontaneous transfer |
| Observed | Government system | Market-system | Market-system | Market-system |
| Theoretical prediction | Government system | Market-system | Government and market mixed initiated system | Market-system |

In the case of *Yiwu*, the governance structures of stages I (generation of land coupon) and II (transaction) match with the transaction attributes, but not that of stage III (the process index being used by the developers). Further specific information please find in Appendix B Table A4. Market-based bidding will minimize the possibility of government rent-seeking in the transaction, and the planned configuration may generate redundant transaction costs, so that the construction land index cannot be allocated to the most effective users. The relation between developers and land subjects would need to take the form of competition instead of planning to obtain the expropriated land.

In *Jinzhai*, the governance structures of first and second stages of homestead replacement match with the transaction attributes, while the third stage will need to be adjusted. Further specific information please find in Appendix B Table A5. The construction index could be transacted in the market though the public trading platform. There still exists the possibility and space to improve the governance performance. The government needs to be responsible for the exorbitant fees of making contacts with other departments, and the coordination of relationships in the stage of index linkage. In addition, it is difficult to deal with the problem of corruption linked to the transaction index. Therefore, the governance structure in this stage needs to be modified and improved. It may need to consider the setting of a transaction platform of property rights at the city or county level, in which, the index can be public and competitively transacted [12,41]. In this way, the value of the construction land index could be prominent, improving the efficiency of resource allocation.

In *Jiangning*, the first stage of transaction does not correspond to the theoretical governance structure, and the searching cost, the cost of signing the contract and maintaining the agreement are still very high. Further specific information please find in Appendix B Table A6. We believe that there is still the possibility of improving the governance performance: the transaction between the households can be accomplished in a small-scale market. A mixed initiated system could be adopted before the transaction between households. The government could set up an intermediary organization and intervene in the homestead transfer appropriately. It would publish the transfer information, establish the market platform of supply and demand, which can effectively decrease the transaction costs, correct market imperfection, and improve the transfer efficiency among households.

## 6. Conclusions and Policy Implications

The goal of this article was to define a suitable classification of governance structures for rural homestead transfer in three different areas in China, in order to guarantee high performance and develop allocation efficiency, to explain its rationality, and to find possible improvement plans to choose the suitable governance structure for each area. Our classification of homesteads transfer was based on the institutional economic logics of transaction costs, and the division into government, market, and mixed systems. We have been able to assess the relative importance of transaction costs, to describe their evolution at different stages of the RHT process, to make comparisons between the different situations, and to establish differences between the actual and predicted governance structures in the three case studies. Actually, there is no optimal type of governance structure. It heavily depends on local circumstances which should be reflected in transactional attributes.

These results open a way for improving governance structures in different areas, as well as different stages. The three cases have reflected that it is important to choose the appropriate governance structure to guarantee the operation performance in the process of RHT, in order to improve the efficiency of resources allocation. (1) It is crucial to choose the corresponding governance structure according to local circumstances and transaction attributes in different areas. (2) The choice of governance structure for the RHT in various places should strictly follow the local transaction attributes, allowing multiple forms of governance structures to exist in different regions, and their optimized combination will help reduce transaction costs and improve the efficiency of resource allocation. (3) The government should play the role of good guidance in the preliminary stage of RHT, in order to decrease the transaction costs. During the market phase, it should retreat from

the process, avoiding misplaces and offside and giving play to the function of service, to develop the allocation efficiency of the rural homestead to a highest level.

Our research sheds light on the developments and vitalization of thousands of Chinese villages, as well as rural areas in the developing world with similar conditions as the studied cases. However, our results need to be verified in extensive parts of rural China and some questions remain unsolved. What is the impact of the government promoted homestead transfer on the household labor migration and accordingly the household well-beings? These research areas have not been further and deeply explored. These may be our following research assignments.

**Author Contributions:** Original draft writing, data collection, paper modification, primarily conceptual framework, Y.Z.; Conceptual framework, paper rewriting and modification, A.T.; paper modification and editing, M.E. All authors have read and agreed to the published version of the manuscript.

**Funding:** We would like to acknowledge the funding of China Scholarship Council, Foundation of China (File No. 201806850079). The APC was funded by Nanjing Agriculture University.

**Conflicts of Interest:** The authors declare no conflict of interest.

## Appendix A

**Table A1.** Basic information on the research areas of *Yiwu*.

| Territory | Geographic Position | Population | Land Use Condition |
|---|---|---|---|
| *Yiwu* | Located in the central part of Zhejiang province. Neighboring *Dongyang* in the east, Wuyi in the south, *Lanxi* in the west, 18 km to *Dongyang* county, surrounded by the mountains in the east, south, and north three sides. | The county administers 8 sub districts, and 6 towns. The registered population of 953,312 has recently increased. The rural population reaches 341,472, while the urban population is 440,748, the urbanization is 76.2%. | The total land acreage reaches 1105.46 km$^2$, with middle and lower mountains, hills, down land, and plains in the territory. |

**Table A2.** Basic information on the research areas of *Jinzhai*.

| Territory | Geographic Position | Population | Land Use Condition |
|---|---|---|---|
| *Jinzhai* | Located in frontier of west Anhui, the hinterland of *Dabie* mountain. It sites in the junction of three provinces, seven counties and two areas, and neighbors with Henan and Hubei provinces in the west and south sides. | The county administers 23 towns and 225 administrative villages. The registered population is 683 thousand, the registered urban population 111 thousand, the urbanization rate of registered population 16.3%. | The total land acreage is 3814 km$^2$; the landscapes are made of middle and low mountain, basins, and valley plain. It is the tourism resource county of the biggest area of Anhui province. |

**Table A3.** Basic information on the research areas of *Jiangning*.

| Territory | Geographic Position | Population | Land Use Condition |
|---|---|---|---|
| *Jiangning* | Located in the south central of Nanjing city. Surrounded the main city of Nanjing from the east, west, and south on three sides. It borders on *Qixia* area and Jurong county in the east, *Lishui* area in the southeast. | The county administers 10 sub districts, 129 communities, and 72 villages. The permanent resident population of urban is 905.2 thousand, which occupy 72.5% of the total population. | Called "60% of mountains, 10 of water and 30 of plains". The total acreage is 1561 km$^2$; the water area 186 km$^2$. The practical control area is 157.3 thousand hectares, the farmland 110.3 thousand hectares. The construction land is 36.5 thousand hectares, the unused land 10.5 thousand hectares. The normal geomorphology is low mountains, hills, down land, plains, and basins |

## Appendix B  Transaction Attributes of Homestead Transfer

**Table A4.** The transaction attributes and the appropriate governance structures in *Yiwu*.

| Transaction Attributes | The Analysis Outcome of Homestead Transaction Case in *Yiwu* | The Generation of "Land Coupon" | The Transaction of "Land Coupon" | The Floor of "Land Coupon" |
|---|---|---|---|---|
| | Heterogeneity of property rights | strong | weak | weak |
| | Local assets specificity | low | | |
| | Scale of transfer | big | big | big |
| Asset specificity | Cultivated land resource condition | low | | |
| | Social security function of homestead | weak | | |
| | Function of property | strong | | |
| | Off-farm work level | high | | |
| Uncertainty | Uncertainty of information acquisition | strong | | |
| | Uncertainty of corruption | high | | low |
| Transaction frequency | Transaction frequency | high | low | low |
| | Transaction costs expectation | +-++++- -++ | -+- | -+- - |
| | Governance structure matching | government system | market system | market system |

Source: Data research. (Note: "+" means high or strong, "0" means middle or moderate level, "-" means low or weaker level. Each sign represents one type of transaction attributes).

**Table A5.** The transaction attributes and the appropriate governance structures in *Jinzhai*.

| Transaction Attributes | The Analysis Outcome of Long-Distance Replacement Case in *Jinzhai* | The Stage of Homestead Reclamation | The Stage of Centralized Residence | The Stage of Index linkage |
|---|---|---|---|---|
| | Heterogeneity of property rights | strong | strong | |
| | Local assets specificity | strong | high | |
| | Scale of transfer | big | big | |
| Asset specificity | Off-farm work level | weak | weak | |
| | Property function | weak | | |
| | Social security function of homestead | strong | | |
| | Cultivated land resource condition | high | | |
| Uncertainty | Uncertainty of information | high | high | high |
| | Uncertainty of corruption | high | high | high |
| Transaction frequency | Transaction frequency | high | | low |
| | Transaction costs expectation | ++++++++- - | +++-++ | ++- |
| | Governance structure matching | government system | Government system | market system |

Source: Data research. (Note: "+" means high or strong, "0" means middle or moderate level, "-" means low or weaker level. Each sign represents one type of transaction attributes).

**Table A6.** The transaction attributes and the appropriate governance structures in *Jiangning*.

| Transaction Attributes | The Analysis Outcome of the RHT Case in *Jiangning* | Before Government Intervention | Government Intervention | Before Spontaneously Transfer | Spontaneously Transfer |
|---|---|---|---|---|---|
| **Asset specificity** | Heterogeneity of property rights | strong | weak | low | |
| | Local assets specificity | moderate | | moderate | moderate |
| | Scale of transfer | big | big | small | |
| | Off-farm work level | moderate | | moderate | moderate |
| | Function of property | moderate | | moderate | |
| | Social security function of homestead | moderate | | moderate | |
| | Cultivated land resource endowment | low | | moderate | |
| **Uncertainty** | Uncertainty of information | high | | strong | weak |
| | Uncertainty of corruption | high | high | | |
| **Transaction frequency** | Transaction frequency | high | low | low | |
| | Transaction costs expectation | +0++++-000 | -0-0000+- | 00- | 0-0 |
| | Governance structure matching | government | market | government | market |

Source: Data research. (Note: "+" means high or strong, "0" means middle or moderate level, "-" means low or weaker level. Each sign represents one type of transaction attributes).

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
