# Peer review of "Governance Structure of Rural Homestead Transfer in China: Government and/or Market?"

_land, doi:10.3390/land10070745_

Round 1

Author Response

Dear reviewer: 

             Thank you very much, we have modified the paper according to your comment. 

Reviewer 2 Report

dear authors,

it was a true pleasure to read your well-researched and well-written paper. I have only one remark to the content - you formulate three hypotheses in section 3.3, but in the discussion you do not confront these with your results. I would like to see the hypotheses discussed, too.

from formal point of view, two small remarks: a) check the text for occasional typos, and b) i suppose the publisher requires different referencing style.

other than that, I believe the paper needs no further improvement and I would be glad to recommend publishing it.

Author Response

Dear reviewer: 

            Thank you very much ! we have modified the paper according to your comments. 

Reviewer 3 Report

Topic is of wide interest outside China, investigating variants techniques for land consolidation and re-allocation. Sometimes the English is unclear (to me as a native English speaker). The different characteristics of the three case studies and their comparability could be better presented. The potential for coupons leading to involuntary displacement and excessive transaction costs or corruption in the process. To put it simply, who gains and who loses from this process? How widespread is the practice, and is there much variability between the case studies? English expression needs improving to reach a wider readership, with more discussion of the practical applications of the technique, and clarification of 'rural homestead transfer' comparison with similar practices in countries outside China.

Author Response

Dear reviewer: 

              Thank you very much ! We have modified the paper according to your comments. 

Round 2

Reviewer 1 Report

The authors considered the suggestions and made necessary changes. The paper was improved in terms of clarity and merits to be published.